

# Differences and relationship in functional movement screen (FMS™) scores and physical fitness in males and female semi-professional soccer players

Ricardo Martín-Moya[1,*], Lorena Rodríguez-García[2], Beatriz Moreno-Vecino[2], Filipe Manuel Clemente[3,4,5], Antonio Liñán González[6] and Francisco Tomás González-Fernández[1,*]

[1] Department of Physical Education and Sports, University of Granada, Granada, Spain
[2] Department of Physical Activity and Sport Sciences, Pontifical University of Comillas, Palma, Spain
[3] Instituto de Telecomunicações, Delegação da Covilhã, Lisboa, Portugal
[4] Escola Superior de Desporto e Lazer, Instituto Politécnico de Viana do Castelo, Viana do Castelo, Portugal
[5] Research Center in Sports Performance, Recreation, Innovation and Technology, Melgaço, Portugal
[6] Department of Nursing, Faculty of Health Sciences, Melilla Campus, University of Granada, Melilla, Spain
* These authors contributed equally to this work.

Corresponding authors
Antonio Liñán González,
antoniolg@ugr.es
Francisco Tomás
González-Fernández,
ftgonzalez@ugr.es

## ABSTRACT

**Background:** Soccer is the world's most popular sport for both men and women. Tests of athletic and functional performance are commonly used to assess physical ability and set performance goals. The Functional Movement Screen (FMS™) is a widely used seven-test battery developed by practitioners to provide interpretable measure of movement quality. The main objective of the present study was twofold, to analyze the relationship between FMS™ results from male and female soccer players and to compare their physical performance in different tests.

**Methods:** A total of twenty-eight semi-professional soccer players: fourteen male (age: 21.29 ± 1.64 years; weight: 70.66 ± 5.29 kg; height: 171.86 ± 5.35 cm; BMI: 20.90 ± 2.22 kg/m$^2$) and fourteen females (age: 20.64 ± 1.98 years; weight: 63.44 ± 5.83 kg; height: 166.21 ± 12.18 cm; BMI: 23.02 ± 2.50 kg/m$^2$) were recruited for this study. A paired sample $t$-test was used for determining differences as a repeated measures analysis. All the participants conducted the following tests: The Functional Movement Test (FMS™), 10-m linear sprint, 5-0-5 COD Test and Yo-Yo Intermittent Recovery Test—Level 1 (YYIRT Level 1).

**Results:** A $t$-test with data from 505 COD (change of direction) test showed significant differences between groups, $p = 0.001$, $d = 1.11$, revealing faster times in male soccer players (2.50 ± 0.19) in respect with female soccer players (2.70 ± 0.17). Crucially, a $t$-test with data from FMS did not reveal significant differences between groups. Multiple regression for $V0_{2max}$ revealed significant effects ($r = 0.55$, $r^2 = 0.30$, adjusted $r^2 = 0.24$, $F = 5.21$, $p = 0.04$ and standard error = 2.20). On the other hand, multiple regression for 10-m sprint showed significant effects ($r = 0.58$, $r^2 = 0.33$, adjusted $r^2 = 0.28$, $F = 5.98$, $p = 0.03$). The impact of these factors on the correlation

between FMS$^{TM}$ scores and physical performance measures can vary among individuals.

**Discussion/Conclusion:** This study demonstrates the necessity of utilizing and applying multiple field-based tests to evaluate the movement and capabilities of physical performance in sports. Crucially, consider individual variations and factors such as training background, fitness level, and sport-specific demands when interpreting the relationship between the FMS$^{TM}$ and physical performance in both sexes.

## INTRODUCTION

Soccer is the world's most popular sport for both men and women, with 1.1 million federation licenses in Spain (*Ministry of Education, Culture and Sport of Spain (MECD), 2023*). Previous literature has noted the importance of effective movement proficiency for safe and effective long-term physical performance in soccer players (*Asgari et al., 2021*; *Kramer et al., 2019*; *Lim, Seo & Kim, 2020*). Efficient movement patterns and good neuromuscular control are critical for optimizing performance and minimizing injury risk, especially during high-demand activities such as landing, cutting, and rapid directional changes that are common in soccer (*Pardos-Mainer et al., 2021*). Commonly, tests are used to assess levels of physical fitness that provide information on musculoskeletal and cardiopulmonary status. However, there is a lack of knowledge on functional patterns. Altered movement patterns as a result of lack of motor control or joint mobility issues, in the presence of unaffected nervous system could be defined as a dysfunctional movement. In this context, it was in the 2000s that several assessment tools were developed, with the Functional Movement Screen (FMS$^{TM}$) developed by *Cook, Burton & Hoogenboom (2006)* as an effective screening instrument used to evaluate asymmetries which result in mobility and stability deficits. It is intended to pinpoint dysfunctional patterns that may increase the risk of injuries. The FMS itself is not an injury-prevention program and has limited capacity to predict new and detect previous injuries (*Dorrel et al., 2018*; *Asgari et al., 2021*). Furthermore, *Kramer et al. (2019)* studied the relationship between the FMS$^{TM}$ and sports performance, founding a significant correlation in the FMS$^{TM}$ parameters and tests such as longitudinal and vertical jump, and agility, indicating that it could be implemented both to anticipate sports injuries and to predict the performance of athletes. Many investigators have taken this into consideration, resulting in a rapid growth of the interest on the links between physical fitness and functional movement assessment in the last two decades. Nevertheless, to the best of our knowledge, there are not enough research investigating this important topic (*Marques et al., 2017*).

Concerning the assessment of sport performance in soccer, the most commonly used physiological and physical fitness assessment includes tests to measure the linear and change of direction (COD) speed, cardiovascular capacity, anaerobic power, lower-body

power and strength, range of motion (ROM), and technical skills (*Slimani & Nikolaidis, 2018*). However, not many options are available to assess movement patterns into daily sport practice. The FMS™ was developed to bridge this gap (*Cook, Burton & Hoogenboom, 2006*; *Cook et al., 2014*; *Kramer et al., 2019*).

One of the most relevant capacities for practice and competition in soccer would be strength, being considered a pillar of joint stabilization of the hip, knee and ankle (*dos Santos Andrade et al., 2017*) and for successful rehabilitation and readaptation after an injury (*Jiménez-Rubio et al., 2019*). In addition, it is important to consider that the structure of the competition has changed in recent decades towards a more dynamic and faster playing style. Examples of this are shorter ball contact times, increasing number of passes in a match, higher player work density and quicker transitions (*dos Santos Andrade et al., 2017*; *Wallace & Norton, 2014*). Therefore, players are required to apply accelerations, jumps, running at medium and maximum speed, sprints or COD in a match, being a sport of an intermittent nature (*Pardos-Mainer et al., 2021*; *Slimani & Nikolaidis, 2018*). Consequently, speed testing has become a standard element of performance assessments and for this reason, some of the more soccer-specific tests used in practice involve linear sprinting over various distances, including acceleration and maximum speed phases. The linear sprint 10-m performance and the COD measured *via* the 5-0-5 agility test are commonly performed (*Kadlubowski et al., 2021*). Cardiovascular fitness, measured through maximal oxygen consumption ($VO_2$max), is also an essential component for performing and recovering efficiently from the high intensity actions that occur during competitions, as well as for remaining in good physical condition until the end of the match. In this regard, the Yo-Yo Intermittent Recovery Test (YYIRTL) Level 1 is one of the regular options to estimate aerobic fitness in soccer players (*Bok & Foster, 2021*).

Finally, in order to complete the evaluation, it would be appropriate to consider the overall condition of the movement patterns, as one of the most effective and direct ways to detect muscle deficits, postural misalignments and lack of joint mobility that have an impact on performance and, especially, injury prevention. The FMS™ is presented as a low-cost, noninvasive, and effective test of assessing essential movement patterns in soccer players (*Kramer et al., 2019*; *Lim, Seo & Kim, 2020*). The main objective of the present study was to analyze the relationship between FMS™ results and physical performance in male and female soccer players.

## MATERIALS AND METHODS

### Participants

A total of twenty-eight semi-professional soccer players: fourteen males (age: 21.29 ± 1.64 years; weight: 70.66 ± 5.29 kg; height: 171.86 ± 5.35 cm; BMI: 20.90 ± 2.22 kg/m$^2$) and fourteen females from one club (age: 20.64 ± 1.98 years; weight: 63.44 ± 5.83 kg; height: 166.21 ± 12.18 cm; BMI: 23.02 ± 2.50 kg/m$^2$) were recruited for this study. To calculate the sample size the following equation was used: Sample Size = Z2 × (p) × (1 − p)/C2, where Z = confidence level (95%); $p$ = 0.05 and C = margin of error 0.05.

The inclusion criteria for this study were (i) they had ≥4 years of competitive experience, (ii) they had not been injured in the last 2 months, (iii) they attended 80% of

the training sessions. If any of the players did not meet any of the inclusion criteria, they could not participate in the study. The training sessions were divided into a warm-up, a main part, and a cool-down. The duration of the sessions was 90 min and they trained 3 days a week plus match day.

The research was conducted according to the ethical principles of the Helsinki declaration for research involving human subjects and was approved by research Ethics Committee of the Pontifical university of Comillas (code: 2021/89). All participants were informed of the main aims of the study and signed the informed consent before the start of the study. Athletes were treated according to the guidelines of the American Psychological Association (APA), to ensure the anonymity of the study participants.

## Measures

### Body mass and height assessment

A SECA 213 model (SECA, Hamburg, Germany) was used to assess body mass and height. During the measurement, participants were barefoot and shirtless. For the measurement of body composition, a TANITA (BC-602-MB) was used, and the subjects were asked to remove their shoes and to stand upright and motionless during the measurement. With the scale there were evaluated the parameters: body weight, % fat mass, % muscle mass, bone mass, BMI, kilocalories, % water and visceral fat.

### The functional movement test (FMS™)

The Functional Movement Test (FMS™) is an assessment tool that attempts to assess the fundamental movement patterns of an individual; a tool of detection such as this offers a different approach to injury prevention and performance predictability. Deficits in mobility and stability of the joints potentially predispose athletes and the average population to an increased risk of injury, as optimal movement patterns can possibly prevent and reduce that risk. The FMS™ is made up of seven fundamental movement patterns that require a balance of mobility and stability. These movement patterns are designed to provide an observable performance of the basic movements of locomotion, manipulation, and stabilization. The tests place the athlete in extreme positions where weaknesses and imbalances become apparent if stability and consistency are not used for adequate mobility (*Cook, Burton & Hoogenboom, 2006*; *Cook et al., 2014*).

The FMS™ test consists of seven fundamental movements patterns to measure movement quality through the FMS test: overhead squat, hurdle step, lunge, shoulder mobility, stability with rotation, leg raise and trunk stability with flexion. Scores range from 0 to 3, with 3 being the best possible score as recommended by *Cook, Burton & Hoogenboom (2006)*, *Cook et al. (2014)*. Three experienced researchers in the administration of the functional movement screen protocol and the performance tests administered all testing on an individual basis. An athlete receives a score of zero if at any time during exercise he or she feels pain anywhere in the body. If pain is present, a score of 0 is awarded and the area of the body that hurts is noted. A score of 1 point is awarded if the person cannot complete the movement pattern or cannot assume the position to perform the movement. A score of 2 points is still awarded if the athlete can complete the

movement but must compensate in some way to perform the fundamental movement. And finally, a score of 3 points is awarded if the person performs the movement correctly without any compensation. Specific comments should be noted defining why a score of 3 points was not obtained. After data collection, the researchers compared the values obtained by each athlete to see if there were discrepancies. If so, the criteria of the majority of them were maintained.

### Physical fitness test

All physical fitness test used in the present study were reliability and validity. For more information see *Clemente et al. (2022)*.

### 10-m linear sprint

The 10-m linear sprint test includes maximum acceleration speed phase. In this study, the 10-m sprint a component of 505 COD test and consists of 10-meter of running split by 5-m of initial sprint, a turn, and 5 meter of final sprint (*Rouissi et al., 2017*).

### Change of direction Test 5-0-5

The 505 COD test is a deceleration-change of direction test. The 505-agility test was conducted as follows:

Change of direction was assessed using the Chronojump-Bosco system® (Barcelona, Spain) photocells developed (*de Blas et al., 2012*). This test mainly measured the athletes' ability to change direction. For data collection, the test had to be performed three times with a 3′ rest between each run. The athlete ran from the 15 m marker to the line and across the 5 m markers, made the turnaround turn after passing the line with both feet and ran back across the 5 m markers. Time was stopped once the athlete passed the 10-m line where the photocells were located for the second time.

### Yo-Yo Intermittent Recovery Test—Level 1 (YYIRT Level 1)

The test was performed following the guidelines of *Gonçalves et al. (2021)*. The YYIRT Level 1 consists of four initial out-and-back runs (from 0 to 160 m) at 10–13 km/h and seven runs (from 160 to 440 m) at 13.5–14 km/h. Subsequently, the running speed continues to increase progressively by 0.5 km/h every eight runs until the participant fails to reach the finish line in time on two occasions. The number of levels finished, and the total distance covered in meters at the end of the test were recorded.

### Procedures

The measurements in the study were taken in two sessions before the start of the training session (from 6 to 8 pm). In order to investigate the relationship between FMS[TM] results and physical performance in male and female soccer players, participants were asked to maintain their daily life routines. In addition, testing was performed between 48–72 h after of match to ensure full recovery. Evaluation sessions were carried out during the season between February and March. In the first session, anthropometric measurements were taken to measure body composition and then the FMS[TM] tests were performed. The tests took place in a private room near the changing rooms with a stable temperature of 22 °C and a relative humidity of 52%. In another session, the fitness assessments took place on a

synthetic turf field with an average temperature of 23.5 °C and a relative humidity of 70 °C ± 3%, and during the assessments there was no wind or rain. Before each assessment, the participants performed a warm-up consisting of different phases: a general activation phase with 5 min of jogging, general and specific joint mobility, dynamic stretching, and a gesture-specific warm-up. The warm-up consisted of general joint mobility, 5 min of jogging at moderate speed and the following self-loading exercises: 15 squats including heel raises at the end of the gesture, 15 anteversion-retroversion movements of the pelvis, 15 external-internal rotation movements of the hip in standing position and 10-monopodial dead weight lifts.

### Statistical procedures

Data are presented as mean ± standard deviation (SD) or percentages. Normal distribution and homogeneity tests (Kolmogorov–Smirnov and Levene's, respectively) were conducted on all metrics. A paired sample $t$-test was used for determining differences as a repeated measures analysis (female soccer players and male soccer players). The correlation was presented using the Pearson correlation coefficient ($r$), and the effect size ($d$) was calculated through Cohen's $d$ (*Cohen, 1992*). The interpretation of the d regardless of the sign, followed the scale: very small (0.01), small (0.20), medium (0.50), large (0.80), very large (1.20), huge (2.0) as initially suggested by Cohen and expanded by *Sawilowsky (2009)*. In addition, multiple regression analysis was used to model the prediction of FMS from the remaining variables. In this regression analysis, all variables were examined separately. Data were analyzed using SPSS v.26 (SPSS Inc., Chicago, IL, US).

## RESULTS

Descriptive statistics were calculated for each variable (Table 1).

A paired sample $t$-test was used for determining differences as a repeated measures analysis (female soccer players and Male soccer players). A $t$-test with data from Yo-Yo Intermittent Recovery Test—Level 1, and $V02_{max}$ estimated showed significant differences between groups, $p = 0.01$, $d = -1.02$. In this sense, dataset revealed higher distances in male soccer players (51.12 ± 2.53 m) in respect with females soccer players (48.36 ± 2.89 m). A $t$-test with data from 505 COD test showed significant differences between groups, $p = 0.001$, $d = 1.11$, revealing faster times in male soccer players (2.50 ± 0.19 s) in respect with females soccer players (2.70 ± 0.17 s). In the same line, another $t$-test with data from 10-m sprint showed significant differences between groups, $p = 0.001$, $d = 1.24$, reflecting faster times in male soccer players (1.77 ± 0.11 s) in respect with females soccer players (1.90 ± 0.10 s). Crucially, a $t$-test with data from FMS did not reveal significant differences between groups (male soccer players (16.29 ± 1.64) and females soccer players (15.86 ± 2.48)). See Table 1 for more information.

Posteriorly, a correlation analysis was performed between $V02_{max}$ and FMS[TM] scores for male soccer players and female soccer players and dataset revealed a large positive significant correlation for male soccer players, $r = 0.55$, $p = 0.04$, but did not reveal significant correlation for females soccer players, $r = 0.13$, $p = 0.65$. Another correlation analysis between FMS[TM] scores and 505 COD test showed a large negative correlation for

**Table 1 Performance variables in both groups (mean ± SD).**

| | Male soccer players ($n = 14$) | | | | Female soccer players ($n = 14$) | | | | | |
|---|---|---|---|---|---|---|---|---|---|---|
| | Mean ± SD | Minimum | Range | Maximum | Mean ± SD | Minimum | Range | Maximum | t-test (p) | Cohen d |
| **FMS (AU)** | 16.29 ± 1.64 | 13.00 | 6.00 | 19.00 | 15.86 ± 2.48 | 10.00 | 10.00 | 20.00 | 0.47 | −0.20 |
| **YYIR1. Distance (m)** | 1,752.86 ± 301.34 | 1,120.00 | 1,120.00 | 2,240.00 | 1,424.29 ± 343.93 | 640.00 | 1,240.00 | 1,880.00 | 0.01* | −1.02 |
| **V02$_{max}$ (ml/kg/min)** | 51.12 ± 2.53 | 45.81 | 9.41 | 55.22 | 48.36 ± 2.89 | 41.78 | 10.42 | 52.19 | 0.01* | −1.02 |
| **505 COD test (s)** | 2.50 ± 0.19 | 2.21 | 0.60 | 2.81 | 2.70 ± 0.17 | 2.49 | 0.64 | 3.13 | 0.001** | 1.11 |
| **10 m (s)** | 1.77 ± 0.11 | 1.60 | 0.32 | 1.92 | 1.90 ± 0.10 | 1.76 | 0.39 | 2.15 | 0.001** | 1.24 |

Note:
AU, arbitrary Unity; YYIR1, Yo-Yo Intermittent Recovery Test—Level 1. Significance level, n.s (no significant); *$p < 0.05$; **$p < 0.01$. Data as mean ± SD.

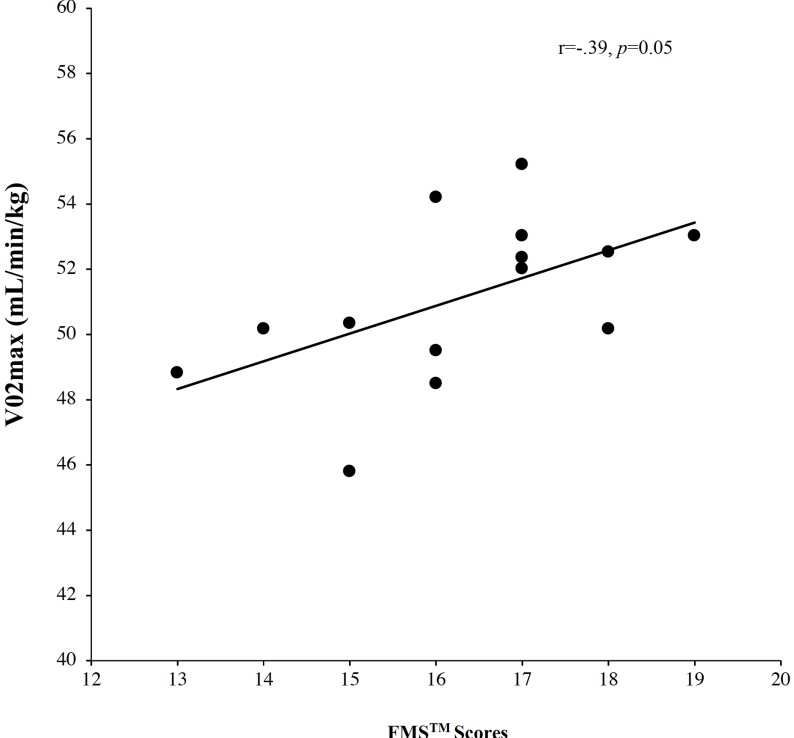

**Figure 1 Correlations analysis between FMS™ scores and V02$_{max}$.**

male soccer players, $r = −0.58$, $p = 0.03$, however did not showed significant correlation for females' soccer players, $r = −0.37$, $p = 0.20$. A correlation analysis between FMS™ scores and 10-m sprint did not reveal significant correlation for both groups, males: $r = −0.18$, $p = 0.53$ and females: $r = −0.28$, $p = 0.34$ (More information in Figs. 1 and 2).

Finally, a multiple regression analysis was performed to verify which fitness values, (agreement with the correlation analysis), could be used to better explain the importance of FMS™ scores. On the one hand, multiple regression for V02$_{max}$ revealed significant effects ($r = 0.55$, $r^2 = 0.30$, adjusted $r^2 = 0.24$, $F = 5.21$, $p = 0.04$ and standard error = 2.20). On the other hand, multiple regression for 10-m sprint showed significant effects ($r = 0.58$, $r^2 = 0.33$, adjusted $r^2 = 0.28$, $F = 5.98$, $p = 0.03$ and standard error = 0.16).

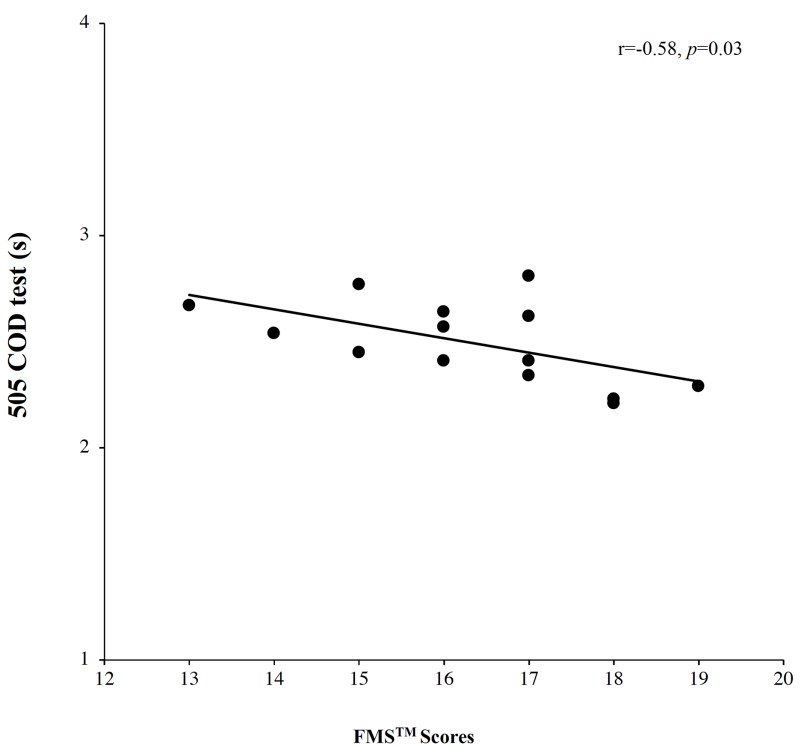

**Figure 2 Correlations analysis between FMS^TM scores and 505 COD.**

## DISCUSSION

The main objective of the present study was to analyze the relationship between FMS results from male and female soccer players and their physical performance in different tests. The FMS is a low-cost, noninvasive, and effective test of assessing essential movement patterns (*Marques et al., 2017*; *Medeiros et al., 2019*). Literature research shows that the FMS is a method that establishes good reliability (*Minick et al., 2010*; *Shultz et al., 2013*) and specificity in detecting athletes who are more likely to suffer a sport injury (*Asgari et al., 2021*; *Serenko & Lafontaine, 2018*).

Males had a superior performance in all tests of physical abilities ($VO_{2max}$, 505 COD and 10-m sprint). Through youth to adulthood, typical reference data indicate that males have a greater propensity for musculoskeletal strength and power than females, therefore, the results of the physical performance tests were expected (*Kramer et al., 2019*; *Lesinski et al., 2020*). In this study, females and males did not differ in FMS^TM scores. Through the literature on the FMS^TM, there is conflicting evidence regarding the presence of sex differences in the total FMS^TM score (*Chimera, Smith & Warren, 2015*; *Paszkewicz, McCarty & Lunen, 2013*). *Domaradzki & Koźlenia (2020)*, demonstrated that males have a higher degree of movement quality patterns than females regarding the total FMS score. However, recent research has demonstrated that females have a higher performance when evaluating composite scores in FMS^TM (*Thomas, Holmes & Wolf, 2023*). Consequently, this study provides another example of sex comparison *via* the composite FMS^TM score, specifically in female and male soccer players.

While there may be general differences in physical performance between males and females due to biological factors such as muscle mass and hormonal variations (*Ansdell et al., 2020*), it is important to recognize that individuals within each gender can have a wide range of abilities and capabilities. Many females can perform exceptionally well on the FMS (*Domaradzki & Koźlenia, 2020*), and there are numerous examples of highly skilled females' athletes across a variety of sports (*Ansdell et al., 2020*; *Korobeynikov et al., 2020*; *Strykalenko et al., 2020*). In this context, no correlations are found between FMS$^{TM}$ and physical performance in females in this study. Research examining the relationship between the FMS$^{TM}$ and physical performance in females has yielded mixed results. Some studies have found weak to moderate correlations between FMS$^{TM}$ scores and measures of physical performance, such as jumping, agility, or speed tests, in females (*Davies et al., 2022*; *Kramer et al., 2019*; *Lockie et al., 2015*). These correlations suggest that individuals with better movement quality on the FMS$^{TM}$ tend to perform better on certain physical performance measures (*Davies et al., 2022*). In this research, two of the female players and one male got FMS scores below 14 points. This score is typically regarded as the cut-off point for a higher risk of injury. In addition, it is important to note that the FMS$^{TM}$ is primarily designed as a screening tool to identify movement dysfunctions or imbalances, rather than being directly linked to specific athletic performance outcomes (*Farley et al., 2020*). These results suggest that female athletes should be able to show a great ROM in different tasks. However, as will be explained, this study also showed that greater ROM as measured by FMS$^{TM}$ was not associated with better physical performance. Therefore, strength and conditioning coaches should be aware that ROM measures obtained from the FMS$^{TM}$ may have limited impact on females' specific sport performance.

Otherwise, men show a relationship between the results in the FMS$^{TM}$ test and VO2 max, this result is one of the findings of the present study that differs from those found in the literature. In this sense, several studies have shown that there isn't a direct correlation or causation established between both in scientific literature (*Fbases, Burden & Hicks, 2021*). However, it is reasonable to speculate that a player with a high VO2 max and good movement quality (as assessed by FMS$^{TM}$) might have an overall advantage in soccer due to a combination of good aerobic fitness and low risk of movement-related injury (*Pfeifer, 2017*). It is worth noting that both FMS$^{TM}$ and VO2 max can be influenced by targeted training, so soccer training programs often incorporate elements designed to improve both movement quality and aerobic fitness (*Yıldız, 2018*).

Likewise, men showed a relationship between FMS$^{TM}$ and the change of direction. The association between FMS$^{TM}$ and agility in males may be attributed to the similar patterns of coordination between the tests (*McGill, 2010*). During maneuvers that require changing directions or completing tasks, the individual is tasked with maintaining their single leg stability and core activation, this is similar to the core activation and single leg stability required during the rotary motion and in-line lunge of the FMS$^{TM}$ (*Prieske et al., 2016*). Additionally, proper core activation is crucial to the fundamental movements in sport (*e.g.*, the agility change of direction test), it is also important in the development and transmission of force through the kinetic chain (*Hung et al., 2019*; *Zemková & Zapletalová, 2022*). Nevertheless, some research suggests that there might not be a direct correlation

between FMS[TM] scores and athletic performance measures like COD (*Bennett et al., 2022*). This is because the FMS[TM] assesses fundamental movement patterns, which are only a part of the complex set of skills required in COD, such as speed, agility, strength, and sport-specific skills.

On the other hand, males did not show any relationship between FMS and 10-m sprint. This could be explained because performance tests may focus on specific skills such as sprinting, jumping, throwing, or sport-specific movements (*Kramer et al., 2019*). These tests are often designed to assess an individual's ability to generate power, speed, or accuracy in a particular activity (*Parr et al., 2020*; *Wagner et al., 2019*). While these tests may require good movement mechanics, they may not necessarily require the same level of range of motion as the FMS[TM]. For instance, through the FMS[TM] deep squat test, a position in which "the femur is under horizontal" is required for a maximum score. In contrast, outcomes in the 10-m sprint or other similar tests are not dependent on the use of a full range of motion.

The empirical results reported herein should be considered in the light of some limitations. The limitations of this study include the total number of participants, it is necessary to expand the sample in order to obtain more significant results. Another limitation is the collection of data at a single moment in time during the season. Future studies assessing FMS[TM] and physical performance measurements at the beginning and end of specific individual training periods to improve movement quality will help to better understand the impact of FMS[TM] improvements on physical performance outcomes, bringing clarity to this topic.

## CONCLUSIONS

This research's findings demonstrate the necessity of utilizing and applying multiple field-based tests to evaluate the movement and capabilities of physical performance in sports. The FMS[TM] focuses on fundamental movement patterns and aims to identify any limitations or asymmetries that may increase the risk of injury or impact overall movement quality. Additionally, it is crucial to consider individual variations and factors such as training background, fitness level, and sport-specific demands when interpreting the relationship between the FMS[TM] and physical performance in both sexes. The impact of these factors on the correlation between FMS[TM] scores and physical performance measures can vary among individuals. Strength and conditioning coaches may find it more valuable to use other methods to assess females' athletic weaknesses.

## ACKNOWLEDGEMENTS

We thank the female and male semi-professionals soccer players and those responsible for the team, for their collaboration and participation in the study.

### Funding

The authors received no funding for this work.

## Competing Interests

The authors declare that they have no competing interests.

## Author Contributions

- Ricardo Martín-Moya conceived and designed the experiments, performed the experiments, prepared figures and/or tables, authored or reviewed drafts of the article, and approved the final draft.
- Lorena Rodríguez-García conceived and designed the experiments, performed the experiments, prepared figures and/or tables, authored or reviewed drafts of the article, and approved the final draft.
- Beatriz Moreno-Vecino conceived and designed the experiments, performed the experiments, prepared figures and/or tables, authored or reviewed drafts of the article, and approved the final draft.
- Filipe Manuel Clemente conceived and designed the experiments, prepared figures and/or tables, authored or reviewed drafts of the article, and approved the final draft.
- Antonio Liñán González conceived and designed the experiments, prepared figures and/or tables, authored or reviewed drafts of the article, and approved the final draft.
- Francisco Tomás González-Fernández conceived and designed the experiments, performed the experiments, analyzed the data, prepared figures and/or tables, authored or reviewed drafts of the article, and approved the final draft.

## Human Ethics

The following information was supplied relating to ethical approvals (*i.e.*, approving body and any reference numbers):

Research Ethics Committee of the Pontifical university of Comillas "Condición Física y fútbol. Evaluación holísitica de jóvenes jugadores de fútbol" (code: 2021/89).

## Data Availability

The dataset with data from 28 players is available in the Supplemental File.

## Supplemental Information

Supplemental information for this article can be found online at http://dx.doi.org/10.7717/peerj.16649#supplemental-information.

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
