# Peer review of "Differences and relationship in functional movement screen (FMS™) scores and physical fitness in males and female semi-professional soccer players"

_PeerJ, doi:10.7717/peerj.16649_

## Round 0.1 · original submission · Major Revisions

Both reviewers raised important points that need to be addressed in major revisions to the manuscript.

Specific considerations should be given to the introduction. In order to emphasize the contribution of the study; the research deficit needs to be clearly stated based on current evidence. Please highlight why another study on FMS and performance in sports is needed.

Also, as Reviewer #2 points out, there are multiple language, grammatical and technical errors, and thorough editing is needed.

**Language Note:** The Academic Editor has identified that the English language must be improved. PeerJ can provide language editing services - please contact us at copyediting@peerj.com for pricing (be sure to provide your manuscript number and title). Alternatively, you should make your own arrangements to improve the language quality and provide details in your response letter. – PeerJ Staff

Reviewer 1 ·

Basic reporting

No comment.

Experimental design

No comment.

Validity of the findings

No comment.

Annotated reviews are not available for download in order to protect the identity of reviewers who chose to remain anonymous.

·

Basic reporting

The article currently contains many language, grammatical and technical errors. It is advised that thorough editing should be done. Please see the comments in the first abstract (without line numbers) as well as the numbered abstract. The abbreviation for the FMS is not used in the same format consistently, e.g., FMSTM vs FMSTM superscript or just FMS. References include good, recent/not dated publications, however, the Reference list should be edited, e.g., lines 399, 403, 419, 423 where the article names are capitalized.

The in-text references should also be checked for consistency regarding the font style used.
Add date to citation in line 166.

Experimental design

The research aim is not clear when first read: "The main objective of the present study was to analyze
the relationship between FMSTM results from male and female soccer players and their physical performance in different tests." Maybe rephrase: "to analyze the relationship between FMSTM results and physical performance in male and female soccer players." The title could also be addressed.

Participants:
I assume all players were from one club.
Do you have a reference for the specific inclusion criteria (line 109)?

Measures:
Any specific instructions to the players before testing? Specifically, because testing took place between 18h00 and 20h00. Any information regarding no exercise on day of testing, caffeine or water ingestion (because you have measured % water), how many days after last match to ensure full recovery?

FMS:
Were the researchers experienced in conducting the FMS? Were any assessments for reliability done?

10-m linear sprint:
It is not correct to state that the 10-m sprint also measures a maximum speed phase (line 152). Sprinters typically reach their highest speed between 50 m and 80 m (or even later in a race) (Healy, 2022; Maćkała, 2015).

Validity of the findings

Did the authors consider reporting oo the FMS scores separately for left and right legs?
Lines 268 - 272: The content of the sentences in this section creates the impression that the authors regard the "flexibility" and "range of motion" as the same concepts or synonyms.

The authors state in Lines 266 and 267: "The FMSTM is primarily designed as a screening tool to identify movement dysfunctions or imbalances, rather than being directly linked to specific athletic performance outcomes."

It could be worthwhile to mention the fact that two of the female players and one male got FMS scores below 14 - the score that is typically regarded as the cut-off point for a higher risk of injury.

---

## Round 0.2 · Minor Revisions

The manuscript was appropriately revised. However, both reviewers requested minor changes that should be considered before acceptance.

Please also address the follwoing points:
- COD should be spelled out the first time it is mentioned in the abstract
- specify "new t-test" and "last t-test" or delete "new / last"
- Commonly, tests are used to
- please split the sentence "...cardiopulmonary status, however, there is a lack of knowledge on functional patterns." in the introduction into two sentences (start the second sentence with however)
- Add "change of direction" the the subheading "Test 5-0-5"
- be consistent with spellings. Sometimes it is "change-of-direction" and sometimes "change of direction"
- "18:00 to 20:00" should be "6 to 8pm"
- add units to all reported data in the results (also those that are in brackets)
- please check the entire manuscript for English grammar

**Language Note:** The Academic Editor has identified that the English language must be improved. PeerJ can provide language editing services - please contact us at copyediting@peerj.com for pricing (be sure to provide your manuscript number and title). Alternatively, you should make your own arrangements to improve the language quality and provide details in your response letter. – PeerJ Staff

Reviewer 1 ·

Basic reporting

I suggest to check the grammar again.

Experimental design

I suggest using passive voice minimally in the methodology section

Validity of the findings

no comment

Annotated reviews are not available for download in order to protect the identity of reviewers who chose to remain anonymous.

·

Basic reporting

Thank you for the addressing the reviewers feedback.

Experimental design

Good and corrected where requested.

Validity of the findings

Good.

Additional comments

Thank you for addressing the feedback from the reviewers.
I last few comments:
Line 89: The font style of the reference differs from the rest of the document.
Line 108: Correct spelling of "performance". It currently is "performe".
Line 150: Rephrase; the sentence is awkward.
Line 217: "In addition, testing was done ...". The sentence currently is written without "testing".
Line 218: Read sentence again and make sure it makes sense.
Lines 338 to 341: My apologies if I have previously missed this long sentence, starting with "Otherwise ...". I suggest you rephrase and maybe have two sentences and not the one long sentence with all the aspects you want to mention.
Line 395: "soccer" is written as "occer".

---

## Round 0.3 · Minor Revisions

Please add “semi-professional” to “A total of twenty-eight soccer players” (A total of twenty-eight semi-professional soccer players) and delete from “fourteen females semi-professional soccer players”. Also change females to female

You stated in your response letter that “New and last have been eliminated, since several t-tests were performed.” But the terms are still in the one abstract version. Please remove

The statistics should be mentioned after the description of tests in the abstract. The “normal distribution and homogeneity tests” are not really needed in the abstract. Consider to delete the sentence

You should have at least one sentence for discussion / conclusion in the abstract

Results: Lines 225; 228 and 230 “female’s soccer players” should be “female soccer players”

Same in line 229 for “males soccer players” (its male…)

Discussion: “females and males did not show a greater success between each other in the FMS” change to “females and males did not differ in the FMS”

---

## Round 0.4 · accepted · Accept

I have read the latest version of the revision. All remarks were addressed appropriately and the article now meets all acceptance criteria. I would like to congratulate the authors on the study